# Qualitative study exploring the feasibility of using medication monitors and a differentiated care approach to support adherence among people receiving TB treatment in South Africa

Rachel Mukora [1,2] Noriah Maraba [1] Catherine Orrell,[3] Lauren Jennings,[3] Pren Naidoo,[4] M Thulani Mbatha,[5] Kavindhran Velen [1] Katherine Fielding [6] Salome Charalambous,[1,2] Candice Maylene Chetty-Makkan [7]

For numbered affiliations see end of article.

**Correspondence to**
Rachel Mukora;
rmukora@auruminstitute.org

## ABSTRACT

**Objectives** The tuberculosis (TB) MATE study evaluated whether a differentiated care approach (DCA) based on tablet-taking data from Wisepill evriMED digital adherence technology could improve TB treatment adherence. The DCA entailed a stepwise increase in adherence support starting from short message service (SMS) to phone calls, followed by home visits and motivational counselling. We explored feasibility of this approach with providers in implementing clinics.

**Design** Between June 2020 and February 2021, in-depth interviews were conducted in the provider's preferred language, audiorecorded, transcribed verbatim and translated. The interview guide included three categories: feasibility, system-level challenges and sustainability of the intervention. We assessed saturation and used thematic analysis.

**Setting** Primary healthcare clinics in three provinces of South Africa.

**Participants** We conducted 25 interviews with 18 staff and 7 stakeholders.

**Results** Three major themes emerged: First, providers were supportive of the intervention being integrated into the TB programme and were eager to be trained on the device as it helped to monitor treatment adherence. Second, there were challenges in the adoption system such as shortage of human resources which could serve as a barrier to information provision once the intervention is scaled up. Healthcare workers reported that some patients received incorrect SMS's due to delays in the system that contributed to distrust. Third, DCA was considered as a key aspect of the intervention by some staff and stakeholders since it allowed for support based on individual needs.

**Conclusions** It was feasible to monitor TB treatment adherence using the evriMED device and DCA. To ensure successful scale-up of the adherence support system, emphasis will need to be placed on ensuring that the device and the network operate optimally and continued support on adhering to treatment which will enable people with TB to take ownership of their treatment journey and help overcome TB-related stigma.

## STRENGTHS AND LIMITATIONS OF THIS STUDY

⇒ We conducted the study in three provinces with different health service characteristics, tuberculosis epidemiology and population characteristics, which allowed us to understand feasibility of implementation from a broader perspective.

⇒ Another strength was that the study was done in a routine setting with limited resources and without the use of incentives to providers to increase uptake.

⇒ One limitation is that some of the interviews were conducted by a member of the study management team, although this was limited to senior stakeholders who were unlikely to feel reluctant to freely express their views and to withhold some of their opinions.

**Trial registration number** Pan African Trial Registry PACTR201902681157721.

## INTRODUCTION

Globally, the treatment success rate for new and relapse drug-sensitive tuberculosis (TB) is 86% while in South Africa (SA) this is much lower at 79%.[1] Among those who are HIV positive with TB, the treatment success rate is 78% and 67% for those previously treated with TB.[1] Treatment failure is often a result of non-adherence to treatment, lost to follow-up or unevaluated outcomes.[2] Typical reasons that people with TB (PWTB) might not adhere to treatment include false perceptions of being cured once they feel better, stigma, forgetfulness, lack of social network support and poor user experience of accessing care at clinics.[3] Traditional methods such as Directly Observed Treatment Short course (DOTs), pill counts and self-report have limitations and have not been shown to

improve treatment adherence in PWTB, especially where DOTs programmes are not designed and implemented appropriately.[4–6] The effectiveness of DOTs is influenced by higher DOTs coverage where cure rates have been found to be higher among those DOTs supporters with fewer patients allocated to them.[5] Digital adherence technology (DAT) including medication monitors may overcome challenges to monitoring TB treatment adherence through remotely documenting dosing patterns of PWTB.[7]

Both globally and in SA, there is limited information on the value of DATs in TB treatment. A study done in KwaZulu-Natal SA among drug resistant TB-HIV coinfected inpatients using an older version of the Wisepill device, the RT2000 3G, showed that feasibility challenges for digital pillboxes may include battery failure, device malfunction and problems related to cellular networks.[8] Previous studies have found that other DAT such as short message service (SMS) reminders for TB treatment were insufficient to improve treatment adherence alone. Barriers included frequent changing of phone numbers and uncertainty whether patients were taking their tablets immediately after they received the SMS or not.[9] Another study done in Uganda for HIV treatment showed that challenges for cellphone-based strategies included: use of shared cell phones, technical failures preventing receipt of SMS texts, electricity outages and changing phone numbers.[10 11] Another DAT known as 99 DOTs also has the potential to improve treatment adherence as the PWTB is required to call a hidden phone number within the blister pack so as to indicate when a dose was taken.[10] However, 99 DOTS is limited in its ability to accurately report if a dose was taken since the PWTB may dial the number without actually taking any medication.[10 12] This challenge has been overcome by DATs methods like Video DOT (VOT) which allow heathcare workers (HCWs) to watch the PWTB taking treatment over video conferencing, hence it is considered a more accurate method to ensure that PWTB adhere to treatment.[13] In addition, VOT can be a source of support to those who lack family and friends to support them during their TB illness.[14] However, VOT requires a smartphone, data plan and sufficient bandwidth making it a less affordable method in low-income and middle-income countries such as SA.[13]

Most DATs have focused on monitoring adherence and few studies have used DATs to differentiate between those patients who are struggling and need individual support, and those who are doing well. Given this gap in knowledge, we evaluated the feasibility of using the Wisepill evriMED DAT to monitor TB treatment adherence while using a differentiated care approach (DCA) with a stepwise increase in adherence support for PWTB in SA.

## METHODS
### Study design and study setting
This qualitative study was embedded within a cluster-randomised trial (CRT) that took place in six clinics in each of three provinces of SA: Gauteng (Ekurhuleni district); Western Cape (Klipfontein and Mitchell's Plain districts) and Kwa-Zulu Natal (eThekwini district), the details of which have been published elsewhere.[15] In the intervention arm of the CRT, PWTB received medication monitors with reminders triggering DCA in response to adherence data uploads, carried out from a central database.[15] The DCA was implemented in a progressive manner depending on the number of doses a participant missed.[15] If one dose was missed, then an SMS reminder was sent to the participant.[15] If a second or third dose was missed, then study staff would make a telephone call to the participant and once the fourth dose was missed then a home visit was conducted during which motivational counselling took place.[15] We describe the feasibility of implementing Wisepill evriMED device and differentiated care from a stakeholder perspective.

### Site selection
There were a total of nine intervention clinics, three in each province. Clinics were selected based on location, HIV prevalence and numbers of patients starting TB treatment per month.[15]

### Study population
The study population consisted of seven purposively selected stakeholders from the Department of Health: one national, one provincial and five district-level representatives, who worked closely with the facilities using the electronic device (Wisepill evriMED DAT). From each intervention facility, we interviewed one facility staff (government employee) and one study staff member. All study staff worked on the project for at least 3 months. Facility staff-initiated patients on TB treatment and monitored patients for their scheduled monthly follow-up visits while study staff offered PWTB the device, followed-up on those who had missed doses and provided motivational and adherence counselling.

### Patient and public involvement
Patients or the public were not involved in the design, or conduct, or reporting, or dissemination plans of our research.

### Conceptual framework and themes explored
Using the feasibility framework suggested by Bowen et al,[16] we developed an in depth interview (IDI) guide (online supplemental appendix 1) that covered the following topics; (1) the relative ease of implementation and operation of the technology within existing health systems, technology infrastructure and supply chain and (2) system level challenges of delivering, sustaining and integrating the intervention into the existing TB programme. We used the framework on the 'Integration and sustainability of interventions into health systems'[17] to organise the data.

### Data collection
We conducted a total of 25 IDIs. The sample was representative of providers involved in the intervention while

also considering saturation where themes usually start to converge after 15 interviews. The IDI guide (Appendix I) was piloted with one stakeholder, two facility and one study staff between January and February 2020. The probing questions were adapted after the pilot. These interviews were conducted face to face, and the data were included in the analysis. Data collection continued between June 2020 and January 2021 with each interview lasting between 45 and 90 min. The interviews were all conducted in the providers preferred language by a female PhD student (RM) and a female study coordinator who was a PhD student with qualitative experience (VM). Both researchers established a prior relationship with the providers enabling them to understand the reasons for the study. The interviews were digitally recorded with the providers consent and transcribed verbatim by trained research assistants (masters students). The transcripts were not returned to the providers for comment or correction. Due to the impact of COVID-19 and the national lockdown that took place, these interviews were conducted virtually over Microsoft Teams. No one else was present in the interviews besides the providers and the researchers and field notes were made during the interviews. Saturation was assessed during data collection through asking the same question in different ways and reviewing a sample of the recordings until no new information was obtainable. No repeat interviews were carried out.

## Data analysis

Thematic analysis was used with deductive and inductive approaches.[18 19] At least 10% of the transcripts were coded inductively by two independent researchers so as to reduce bias and improve reliability.[18] A codebook of emerging themes was developed, guided by the framework on the 'Integration and sustainability of interventions into health systems',[17] with discrepancies being resolved through discussion. Where there was no inter-rater agreement, the theme was dropped.[18] The final codebook was used to code the remaining transcripts and any new codes that emerged were also included in this codebook. MAXQDA qualitative software was used for the coding process. We highlight the major themes (overarching theme) and subthemes (specific themes) that emerged and use supportive direct quotations from providers. Providers did not provide feedback on the findings.

## RESULTS

We conducted 25 interviews in total with 18 staff (9 facility staff and 9 study staff) and 7 stakeholders (1 national,1 provincial and 5 at district level) across 3 provinces. Majority of the providers (23 out of 25) were female. None of the approached providers declined to participate. Major themes included (1) Providers were supportive of the intervention, (2) Intervention challenges within the adoption system and broad context, and (3) Ensuring intervention sustainability through constant training of staff and education of PWTB

## Major theme 1: providers were supportive of the intervention

There was buy-in from staff who felt involved and were supportive of the intervention. Stakeholders were also very supportive of the intervention despite their limited involvement.

> … to me it was a very good idea and it was working very well … it improved our cure rate because we had a lot of patients who were defaulting and who and to be referred to the hospital because they developed multi, MDR. So since we had …this intervention, at least we had less patients who developed multi-drug resistance. - FFS_019_7

> …I think…it's magnificent. I think it's… a tool box, it's really a way for us to see what's really happening at the point of … TB treatment, where the patient take… the medication when they open the box. So I think it's really innovation that can be used. … I think it also is a reminder for the patient. You know? That … "I need to take my medication". And it's a way to …to ensure the quality of the programme, that patients are adhering to the … medication…I think it is a critical intervention. – FDS_WC_008 (District level)

Both facility and study staff found the device easy to learn and facility staff who had received a briefing on the study were eager to receive the complete training to support the study staff.

> Because sometimes she (referring to study staff) [ might not be here and it happens that she's not here, either she is sick or she's gone on holiday for December. She says: "…we can show you and then you do this things when we are not here". I said: "… it will be easy… As long as you show us, we can do that device. - FFS_762_003

The staff described the activities of the intervention such as issuing devices, phone calls and making home visits as being well integrated into the TB programme. Providers had strong support for the device for two main reasons that we classified as subthemes in our analysis:

## Subtheme 1: device as a useful reminder and early notification tool

Most providers found the alarm that was fitted on the device to be a useful reminder to patients to take their treatment on time and to attend their clinic visits. The device also alerted providers to PWTB who were not using the device and not taking treatment.

> I feel like it assists with the real time monitoring because now if you check on the system, you would be able to check instead of waiting like someone waiting for their appointment that is months after, you can actually check now on the system that this person there's no activity, let me call and you would actually find that the person has died or maybe the person has—is in hospital is admitted.- FSS_762_006

So, it will help us because …sometimes we notice the other patient…didn't take their medication and start to recall all of them and then the other just default … With the box, someone is getting … a notification that this person is not taking her treatment… - FFS_524_006

However, some staff distrusted that opening of the device meant that PWTB took their pills.

…some patients do that when they open the pill box and … show an adherence of 100%, but still they… have not been taking their medications. So we would try to motivate them, counsel them, explain the disadvantages of not of them not taking the medication. - FSS_519_001

I think because you know patients always find a way to find loopholes within the health system, so at some point they will understand that you're recording the opening and the closing of the box not necessarily them actively taking the medications. So they can easily open and close the box without really taking the medication. - FDS_GP_008 (District level)

### Subtheme 2: DCA allows for patient-centred care

Staff mentioned that they found DCA to be a positive and unique aspect of our intervention.

So, you are no longer administering intervention for one. You sort of trying to be specific …to a person and offer them care in their in their specific sort of situations. So, I think that is a positive thing because you don't assume everyone is the same. You don't assume everyone's situation is the same. You understand that you are dealing with … individuals. That's … what I think is positive about this differentiated model of care. - MSS_732_005

… yah, I think I think the differentiated model of care … is a very positive thing, that it works to an extent because it's not a blanket approach. You don't, you don't think of every patient as the same…you sort of attend to each person in their context and … try to understand what they are going through. - MSS_732_005

The staff also felt that the counselling they provided as part of the DCA helped them to understand their patients' reasons for not taking their treatment thus allowing them to manage them better as they supported them on their treatment journey.

This whole approach I feel like it's a great initiative because it assists in managing the patient—like fully. Like if I can say wholly, not just managing the patient, taking the medication. You could also like… with the counselling you actually find that the problem is that the participant is facing beyond treatment intake.- FSS_762_006

Phone calls to PWTB allowed for clarification on any issues that may not have been clear to the patient while at the clinic.

…he didn't understand but when I called him and I was speaking to him over the phone to understand how they're taking the treatment, I find that … yeah there was mistake … one of the best things about this intervention because some of them are able to even call me up, I don't always now have to call patients. - MSS_732_005

Home visits allowed the staff to have better interaction with their patients and to build trust with them making them feel like someone cares about them.

It's not bad. It's nice to do home visits. It's whereby … you contact with the patients, know each other … where the patient has got the problem, it's with the … problem with the box or she wanted to ask me something that she forget to ask at the clinic. - FSS_522_002

… I think the participant at least get that thing when you do home visits, or you phone them when they are not taking the treatment they feel as if they are cared of, someone is care is cared for me when I am not taking the treatment because I would get a call, or I will get a home visit'- FSS_524_003

Some providers described DCA as more of a journey of supporting the patients and helping build trust between the staff and the PWTB.

… that is really good part of the intervention. I think it … continue to build that trust…And that support for the patients … so I- I think that was impressive. - FDS_WC_008 (District level)

I thought it was only about the box… it's more than them just using the box. It's their treatment journey and being supported to adhere to the treatment and get healed from the TB. - FSS_762_006

### Major theme 2: intervention challenges within the adoption system and broad context of SA

Some staff felt that the trust they built with the PWTB could be threatened by human resource (HR) shortages and a dedicated cadre was needed to ensure successful implementation.

However, when we find at once the research study stop and we over onto implementation, … the constraints is always HR resources and at the operational level, if that initial education isn't framed correctly. There could be a misunderstanding about the purpose of the box. - FDS_WCR_008 (District level)

Challenges that were encountered with the device or the network resulted in delays in the system updating that the box was opened which staff felt created trust issues between them and the participant.

Now it's creating a trust issue between you and the patient, because you phoning regarding treatment that was not taken because … Wisepill says treatment was not taken, while the patient on the other side did take the medication.- FSS_516_002

HCWs reported that some patients received incorrect SMS's due to delays in the system.

Especially when it has been a weekend and then they will not … send out the data of… the weekend like … if there were any missed or any … intakes. Sometimes they will show that they missed the weekend and they did not take the medication, but in three days down the line… the correct information will show. That's where we will see that there were no missed doses.- FSS_519_001

Staff also felt that that the device would not work for some group of patients such as those who abuse substances.

The patient that are on substance abuse, yoh! You know, when they come here, they are ok. At a later stage, once they've started treatment, you realise that this patient is using substances…With those patients, they the device- ya. It's not good. Won't- won't work. Won't- won't help. - FFS_762_003

He took the evriMED box device and the TB meds. That was the last time we saw him on the clinic. He run away from home apparently. He was on drugs. But then…he is back. He's here in the TB room. He's got TB again. So I was asking him the other day: "are you still interested in joining this study?". And he was like: "you know what I did with the first box? I sold it. - FSS_521_001

### Subtheme 1: Provider perceptions of stigma related to use of the evriMED device and DCA

Staff perceived that some of the features of the box such as the alarm and the size of the box may have been a concern to some PWTB in trying to conceal their TB status so they would opt to leave their devices behind when travelling on holiday or going to work for fear of disclosure of their TB status.

*The thing that is common, especially in December … an example is person supposed to take it at 8, they take it at 8 but they not taking it at 8 from the box… I think they went for holidays … This person is taking treatment and the putting it… in a purse. Whereby you'll take in the morning without this alarm ringing.'- MSS_019_002*

So some will come to us and say: "what if the pill box doesn't make so much noise? It will be easier for me to carry around—or if it was a bit … smaller, then it will be easier for me to carry it. But now it's big, I don't want … people seeing me like with this pill box in the taxi or at work.- FSS_519_001

I haven't disclosed to my partner that I'm on TB medication and imagine if I had to carry this box and the

box would be ringing in the morning and it would cause unnecessary fights. - FSS_762_006

Providers perceived that stigma was more of an issue outside the PWTB household as often they would not disclose their TB status to members outside their homes.

So, I noticed that it's comfortable for them to use this box when they are at home with the people that actually know what is going on with them but when they are going to other people that are not aware that they are sick or anything, it seems as if they don't want to carry the boxes. - FSS_762_006

Staff perceived that stigma related to DCA existed in the community and some of them witnessed the PWTB fear of being stigmatised when conducting home visits. PWTB would often ask staff not to wear their uniforms when they visit their homes.

Uhm they hate it when we wear our TB MATE t-shirts, because they are saying: "due to stigma". When we go visit their homes, we must not wear something that will be written 'TB' or 'HIV'. So we wear our normal clothes…- FSS_521_001

Then I took off my jacket, and then the participant saw that TB MATE on my t-shirt. And then the patient told me that I must wear my jacket because what if someone walks in and sees the TB MATE on my t-shirt. So I wasn't aware that the participant didn't disclose… She told me that she's got fear that—because she will be judged because she was drinking alcohol a lot. But now she's got TB. They will say she must go and stay in the back room.- FSS_521_001

Staff perceived that some stigma could have been internalised and not actually experienced by PWTB.

It's their thoughts because the person would feel like it did not want to be seen carrying the box. Because people will think- so he sees all their thoughts. It's not something that really happened.- FFS_732_001

But in the clinic, there wasn't really anything that contributed to the stigma but maybe with now giving the pill box… some of the patients would say that maybe it's gonna show. Maybe with their ARV's they could take it in private. Now we have them having this pill box that is gonna ring even, that is gonna beep and remind them to take treatment… - FSS_732_005

### Major theme 3: ensuring intervention sustainability through constant training of staff and education of PWTB

Education on the intervention emerged as one of the major themes that stakeholders and staff placed emphasis on as a means of managing their patients in a holistic way leading to successful treatment outcomes. Education and training should emphasise the importance of adhering to treatment and should be strengthened at various levels—with the PWTB themselves, those who would be involved

in implementation such as community health workers and also to the community at large.

> My experience was that as the community we need to need to educate the community the importance of complying to medication…we need to educate and educate and educate.- FFS_019_007

> But I definitely think it's… strengthening that interface between the patient and the staff… it's an important interface. And- and of course that too- the 'how' of- of the contact, if it is about 'why you didn't-why you aren't taking your treatment' rather than saying 'how are you doing'- you know- 'are you ok', uhm 'how can I help you'. When we start off with those different conversations, it can lead to different outcomes. - FDS_WC_008 (District level)

The staff felt that constant education and reinforcement would ultimately lead to behaviour change and treatment adherence.

> I think the intervention even though we are the one offering the intervention but we sort of gave patients the power to understand the… reasons why they are taking treatment, why is it important for them to…adhere to the treatment…we were actually able to make patients to take their treatment journey into their own hands … it's almost like we were empowering patients…- MSS_732_005

## DISCUSSION

Using the Wisepill evriMED device to support adherence to TB treatment appears to be a feasible option in SA. Providers were supportive of the evriMED device and the DCA, as they felt it had a positive impact on the TB patient's programme. Stakeholders were supportive despite their involvement being limited to approval of the study and providing oversight of the TB programme.

This study has shown that DATs is an innovative approach that might improve the management of PWTB by allowing individual differentiated support. Early notification of missed doses from the device allowed staff to intervene through SMS messaging and phone calls, thus reducing the need for a home visit. This freed up time for staff to focus on other duties. These findings are similar to other studies where differentiated care models used in HIV treatment delivery have addressed challenges such as overcrowded facilities, overburdened staff and long waiting times in countries like Uganda, Swaziland, Mozambique and SA.[7 20–22]

Staff noted that the differentiated care allowed a journey of educating and supporting PWTB. Through adherence and motivational counselling, staff empowered patients to understand the importance of adhering to treatment and to empower them to take their treatment journey into their own hands. For scale-up, the real time monitoring feature of the device is important so that the correct individuals are promptly identified for further support before they become lost to follow-up. This also allows for the efficient utilisation of resources to those in most need, as seen in various studies.[6 23 24] More attention should also be given to relationship training as the sustainability of the intervention depends on a sound provider–patient relationship. Correct framing of the intervention as a support tool would also help ensure that the intervention is well perceived once scaled up. Training needs to be constant and not once-off to reinforce their understanding of the intervention. In a routine TB programme setting, training of healthcare providers could be conducted by subdistrict co-ordinators since they visit facilities on a regular basis for monitoring purposes.

Some challenges with the technology were cited such as alarm malfunction, incorrectly sent SMS's and short battery lifespan. These challenges created distrust from PWTB, and the device would, therefore, need to be of the highest quality to ensure sustainability of the intervention. A systematic review of DAT for the management of TB therapy found similar feasibility challenges remain in low-income and middle-income settings.[10] Once the intervention is scaled up, the shortage of HR could serve as a barrier to ensuring that providers receive the correct information. For successful scale-up, some staff recommended that a dedicated cadre should be in place to follow up PWTB to ensure that they understand the importance of taking treatment. However, since the support system had reduced the need for home visits and follow-up calls, the current TB cadres, clerical staff and community-based teams were also seen as being sufficient. Despite any challenges experienced within the adoption system and broad context, the staff were all very supportive of the intervention and even recommended that it should be used for all TB and chronic patients as well as in other facilities.

Fear of stigma and the disclosure of one's TB status has the potential to act as a barrier to scale up and sustainability of the intervention, if not well addressed. Stigma and fear of disclosure resulting in lower acceptance of DAT technologies among multidrug resistant TB patients have also been reported in India,[25] with improvements recommended on the design. Disease-related stigma may be more difficult to address, and screening should be used to identify upfront those patients whom stigma and fear of disclosure may limit the use of the MERM.[25]

A limitation of the study was that interviews had to be conducted virtually on Microsoft teams, due to the COVID-19 pandemic and to allow for effective social distancing for both the researchers and the study providers. This new mode of conducting interviews had setbacks such as network and connectivity challenges which interrupted the flow of some of the interviews. The interviewers mitigated this challenge through reiterating what the participant had said to ensure that the correct message had been captured. Through the consenting process, we were able to ensure confidentiality and that

the participant was relaxed. A second limitation is that some of the interviews were conducted by a member of the study management team, although this was limited to senior stakeholders who were unlikely to feel reluctant to freely express their views and to withhold some of their opinions. Patients were not included in this study which is a limitation since their views would have added a broader understanding on feasibility. We conducted the study in three provinces with different health service characteristics, TB epidemiology and population characteristics which allowed us to understand feasibility of implementation from a broader perspective. Another strength was that the study was done in a routine setting with limited resources and without the use of incentives to providers to increase uptake.

## CONCLUSION

DAT such as medication monitors has a huge potential to improve TB treatment adherence. However, to ensure successful scale-up and sustainability of the intervention, the DCA should be used as a platform to constantly educate PWTB and the community at large on TB and the importance of adhering to treatment since technology on its own will not solve treatment adherence issues especially those related to stigma and lack of support. Sound communication between PWTB and providers serves as a key tool to improving treatment outcomes thus more attention should be given to relationship training to ensure that the provider–patient relationships provide the necessary support to those who need it most.

### Author affiliations
[1]The Aurum Institute, Implementation Research Division, Johannesburg, South Africa
[2]University of Witwatersrand, School of Public Health, Johannesburg, South Africa
[3]Desmond Tutu HIV Foundation, Institute of Infectious Disease and Molecular Medicine and the Department of Medicine, University of Cape Town, Cape Town, South Africa
[4]University of Stellenbosch, Stellenbosch, South Africa
[5]Interactive Research and Development, Johannesburg, South Africa
[6]London School of Hygiene & Tropical Medicine, TB Centre, London, UK
[7]Health Economics and Epidemiology Research Office, Faculty of Health Sciences, University of the Witwatersrand, Johannesburg, South Africa

**Acknowledgements** We would like to thank the following: Ekurhuleni, City of Cape Town, eThekwinidistricts and the Ekurhuleni Health District Research Committee (EHDRC) for allowing us to conduct the study in their districts. The study coordinators from Gauteng, Western Cape and KwaZulu-Natal in South Africa for their assistance with data collection namely Israel Rabothata, Vuyelwa Mehlomakulu (VM), Vumile Gumede and Bongani Zondiand their field-based teams of research assistants and interns; all the transcribers who dedicated time to capture the conversations verbatim; all the providers for taking part in this study.

**Contributors** RM, NM, LJ, MTM, KLF, SC and CMCM conceptualised the study. RM and CMCM analysed the data. RM, NM, CO, SC and CMCM interpreted the data. RM wrote the original draft and revised subsequent versions of the manuscript and is responsible for the overall content as the guarantor. SC and CMCM provided guidance on earlier versions of the manuscript. NM, CO, LJ, PN, MTM, KV, KLF, SC and CMCM reviewed and edited previous versions. RM, SC and CMCM are responsible for the overall content of the manuscript. The authors read and approved the final manuscript.

**Funding** The study is funded by (1) Bill & Melinda Gates Foundation (OPP1205388), (2) TB REACH Wave 6 project of Stop TB Partnership (STBP/

TBREACH/GSA/W6-34) and (3) South African Medical Research Council through the Strategic Health Innovation Partnerships.

**Disclaimer** Funders did not play a part in the design of the study or the decision to submit the manuscript for publication.

**Competing interests** None declared.

**Patient and public involvement** Patients and/or the public were not involved in the design, or conduct, or reporting, or dissemination plans of this research.

**Patient consent for publication** Not applicable.

**Ethics approval** The parent study received ethics approval from the three district and provincial ethics committees where the trial sites were located. This qualitative study was approved by the Human Research Ethics Committee (HREC) of the University of Witwatersrand (Ref 180705) and the University of Cape Town (Ref 452/2018). We obtained written informed consent for study participation and permission for digital recording from all providers. Providers were not reimbursed. Participants gave informed consent to participate in the study before taking part.

**Provenance and peer review** Not commissioned; externally peer reviewed.

**Data availability statement** No data are available.

**ORCID iDs**
Rachel Mukora http://orcid.org/0000-0003-4304-1729
Noriah Maraba http://orcid.org/0000-0002-3974-5961
Kavindhran Velen http://orcid.org/0000-0001-8577-3915
Katherine Fielding http://orcid.org/0000-0002-6524-3754
Candice Maylene Chetty-Makkan http://orcid.org/0000-0001-9292-9586

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
