## [Reviewer comments · BMJ Open]

ARTICLE DETAILS

TITLE (PROVISIONAL)	A qualitative study exploring the feasibility of using medication monitors and a differentiated care approach to support adherence among people receiving TB treatment in South Africa
AUTHORS	Mukora, Rachel; Maraba, Noriah; Orrell, Catherine; Jennings, Lauren; Naidoo, P.; Mbatha, M.; Velen, Kavindhran; Fielding, Katherine; Charalambous, Salome; Chetty-Makkan, Candice

VERSION 1 – REVIEW

REVIEWER	Juliet Sekandi University of Georgia, Epidemiology and Biostatistics
REVIEW RETURNED	07-Oct-2022

GENERAL COMMENTS	The authors should elaborate more on how the findings of this study fits in the broader context of other DATs e.g VOT and 99DOTS, how similar or different are these findings especially in LMIC settings also compared to high income settings? Also, from the discussion in relation to this statement 'Training needs to be constant and not once-off to reinforce their understanding of the intervention' please elaborate on how the constant training of health providers and PWTB could be done in a routine TB program clinic. Dr. Sekandi
--

REVIEWER	Richard Garfein University of California, San Diego
REVIEW RETURNED	12-Jan-2023

GENERAL COMMENTS	This qualitative research study examined the feasibility of using the WisePill medication adherence monitoring device to inform a differentiated care approach to supporting adherence among patients receiving anti-tuberculosis treatment in South Africa. Overall, the study design is appropriate and the conclusions are supported by the findings. There are no major flaws in the paper; however, the follow weaknesses should be considered prior to accepting this paper for publication.  1. The paper could use an additional review for grammar and punctuation. For example, the first sentence in the Introduction is overly complex. The points could be made more clear by splitting it into two sentences. In addition, the sentence starting on line 97 should use commas instead of semicolons to separate the items in the list. And, on line 145, add "on" between "based location". 2. Line 100-102 - the references cited are inadequate to support the
---

	statement that directly observed therapy does not improve treatment adherence. At the very least, the authors should cite review articles that do a better job of covering this complex issue than the two small studies cited. Furthermore, the authors should make it clear that the effectiveness of DOTS depends greatly on how the programs are designed and implemented (i.e., not all DOTS programs are created equal). 3. Line 102-107 - The WHO endorsement of digital adherence technology (DAT) was not based on a single study from China. The references for this statement need revision. 4. Line 158 - patients were not included in the study, which is unfortunate. Consider mentioning this in the limitations. 5. Line 198-199 - clarify the distinction between "major" and "soft" themes. Also, consider revising the labeling of these categories in the Results, which are referred to as "Main Theme" and "Major Theme". It is unclear based on the labeling if Sub-Theme 3 is a subset of Major Theme 2 or an extension of Sub-Themes 1 and 2. 6. Line 353 - The issue of substance use among patients doesn't seem to fit well in the theme involving problems staff were having with the WisePill technology. Clarify this connection by potentially revising the description of the theme.
--	---

VERSION 1 – AUTHOR RESPONSE

Reviewer: 1

Dr. Juliet Sekandi, University of Georgia

1. The authors should elaborate more on how the findings of this study fits in the broader context of other DATs e.g VOT and 99DOTS, how similar or different are these findings especially in LMIC settings also compared to high income settings?

We have now included this information in the introduction section on lines 107-117.

2. Also, from the discussion in relation to this statement 'Training needs to be constant and not once-off to reinforce their understanding of the intervention' please elaborate on how the constant training of health providers and PWTB could be done in a routine TB program clinic.

We have added lines 477- 479 to elaborate on how constant training of health providers could be done within the routine TB program.

“In a routine TB program setting, training of healthcare providers could be conducted by sub-district co-ordinators since they often visit facilities for monitoring visits on a regular basis.”

Reviewer: 2

Prof. Richard Garfein, University of California, San Diego

This qualitative research study examined the feasibility of using the WisePill medication adherence monitoring device to inform a differentiated care approach to supporting adherence among patients receiving anti-tuberculosis treatment in South Africa. Overall, the study design is appropriate and the conclusions are supported by the findings. There are no major flaws in the paper; however, the follow weaknesses should be considered prior to accepting this paper for publication.

1. The paper could use an additional review for grammar and punctuation. For example, the first sentence in the Introduction is overly complex. The points could be made more clear by splitting it into two sentences. In addition, the sentence starting on line 97 should use commas instead of semicolons to separate the items in the list. And, on line 145, add "on" between "based location".

Thank you for the comment. We have now split the first sentence in the introduction into two. We have removed the semicolons and used commas in the sentence on lines 85-88. We have also added "on" between "based location" (line 143).

2. Line 100-102 - the references cited are inadequate to support the statement that directly observed therapy does not improve treatment adherence. At the very least, the authors should cite review articles that do a better job of covering this complex issue than the two small studies cited. Furthermore, the authors should make it clear that the effectiveness of DOTS depends greatly on how the programs are designed and implemented (i.e., not all DOTS programs are created equal).

We have added the following reference to support the statement that DOTs does not improve treatment adherence and to make it clear that the effectiveness of DOTs depends on how programs are implemented.

Ntshanga, Sbongile & Rustomjee, Roxana & Mabaso, Musawenkosi. (2009). Evaluation of directly observed therapy for tuberculosis in KwaZulu-Natal, South Africa. *Transactions of the Royal Society of Tropical Medicine and Hygiene*. 103. 571-4. 10.1016/j.trstmh.2009.03.021.

3. Line 102-107 - The WHO endorsement of digital adherence technology (DAT) was not based on a single study from China. The references for this statement need revision.

We have now removed the statement on the WHO endorsement.

4. Line 158 - patients were not included in the study, which is unfortunate. Consider mentioning this in the limitations.

We have included this suggestion on lines 513-515.

5. Line 198-199 - clarify the distinction between "major" and "soft" themes. Also, consider revising the labeling of these categories in the Results, which are referred to as "Main Theme" and "Major Theme". It is unclear based on the labeling if Sub-Theme 3 is a subset of Major Theme 2 or an extension of Sub-Themes 1 and 2.

We have also edited the term "soft theme" to "sub-theme" for consistency.

On lines 196-197, we have clarified the distinction between "major themes" as the overarching theme and "sub-themes" as the more specific theme.

We have corrected the category "main theme" into "major theme".

Sub-Theme 3 is a subset of Major Theme 2 and we have now re-labelled it as Sub-Theme 1 for clarity.

6. Line 353 - The issue of substance use among patients doesn't seem to fit well in the theme involving problems staff were having with the WisePill technology. Clarify this connection by potentially revising the description of the theme.

Thank you for your comment. The description of the theme refers to the intervention (which includes both the Wisepill technology and the Differentiated Care Approach) and the challenges within the adoption system and broad context. The issue of substance use has to do with the broad context of South Africa. We have revised the description of the theme to make this clearer as follows:

“Intervention challenges within the adoption system and broad context of South Africa”

VERSION 2 – REVIEW

REVIEWER	Richard Garfein University of California, San Diego
REVIEW RETURNED	11-Feb-2023
GENERAL COMMENTS	The revised version of this paper is much improved. I have no additional comments.